# Effect of *Muntingia calabura* L. Stem Bark Extracts on Uric Acid Concentration and Renal Histopathology in Diabetic Rats

**DOI:** 10.3390/medicina55100695

**Published:** 2019-10-16

**Authors:** Safrida Safrida, Mustafa Sabri

**Affiliations:** 1Department of Biology Education, Faculty of Teacher Training and Education, Syiah Kuala University Darussalam, Banda Aceh 23111, Indonesia; 2Laboratory of Anatomy, Faculty of Veterinary Medicine, Syiah Kuala University, Darussalam, Banda Aceh 23111, Indonesia, mustafa_sabriyosa@unsyiah.ac.id

**Keywords:** uric acid levels, aloxan, diabetic rat, *Muntingia calabura* L.

## Abstract

*Background and objectives:* This study was designed to determine uric acid concentration and renal histopathology of *Muntingia calabura* L. stem bark extract in diabetic rats and to compare the natural product of *M. calabura* L. stem bark extract with allopurinol. *Materials and Methods:* A completely randomized design was used for the experiment, which consisted of six treatment groups, each consisting of four rats, as follows: 1) NR, normal rat; 2) KN, diabetic rat (negative control); 3) KP, diabetic rats given allopurinol 10 mg/kg body weight; 4) EM150, diabetic rats given the test extract 150 mg/kg body weight/day; 5) EM300, diabetic rats given the test extract 300 mg/kg body weight/day; and 6) EM450, diabetic rats given for extract 450 mg/kg body weight/ day. *Results:* The results showed that *M. calabura* L. stem bark extract decreased (*p* < 0.05) uric acid concentrations in diabetic rats and no specific damage to renal proximal tubular cells was seen. *Conclusions:* It was concluded that *M. calabura* L. stem bark extract has a potential as an antihyperuricemic in diabetic rats. The recommended dose was 300 mg/kg body weight to provide a significant effect on reducing the uric acid level in diabetic rats. Our results support the use of this plant for the treatment of degenerative and inflammatory diseases.

## 1. Introduction

Diabetes mellitus is a high glycemic marker of metabolic disorders, which is a risk factor for cardiovascular disease [1]. In diabetic patients, hypertension and decreased renal function with hyperuricemic are major problems [2]. Hyperurecemia is characterized by high blood uric acid levels due to impaired uric acid metabolism, which puts individuals at risk of developing gout, cardiovascular diseases, hypertension, and metabolic syndrome [3,4]. Since diabetes is often complicated by hypertension and hyperuricemic, efficient therapeutic strategy against these two complications is very important in the treatment of diabetes. A study showed decreased serum uric acid level in patients with diabetes [5]. Hyperuricemia is a risk factor for the pathogenesis of vascular endothelial dysfunction, as well as the involvement of signaling mechanisms that influence the pathogenesis, and this association has been attributed to an inflammatory effect [6,7]. Antihyperuricemic drug can be further explored, and not only as antigout therapeutics [8]. Diabetic rats have intracardiac cytokine protein expression profiles that reflect weaker cardiac defenses compared with healthy rats [9].

Currently, hyperuricemia is treated by using natural ingredients from the *Muntingia calabura* L. plant, a flowering plant in the province of Aceh. It contains polyphenols, flavonoids, ascorbic acid, α tocopherols, and triterpenoids [10]. The leaves of the cherry plant (*Muntingia calabura* L.) are reported to have anti-inflammatory and antioxidant properties [11]. An antioxidant has the ability to decrease blood sugar level [12]. An antioxidant could reduce serum uric acid level [13]. Normally, the use of synthetic drugs cause kidney organ disorder; this study evaluates the effect of the natural ingredients of *M. calabura* on kidney histopatology. The antihyperuricemic potential of the ethanol extract of *M. calabura* L. stem bark from Aceh in diabetic rats has not yet been reported.

The aim of this study was to determine the potential of *M. calabura* L. stem bark extracts from Aceh, an antihyperuricemic, in diabetic rats and to compare the natural *M. calabura* L. stem bark extract with allopurinol.

## 2. Materials and Methods

### 2.1. Preparation of Ethanol Extract of M. calabura L.

*M. calabura* L. stem bark was collected in the mountainous region Aceh Besar of Aceh province, Indonesia. The plant was authenticated by the Department of Biology Education, Syiah Kuala University. The dried plant stem bark material (5 kg) was extracted three times with 96% ethanol at room temperature for 3 days. The solution was filtered and combined, and the organic solvent was removed under reduced pressure and evaporated at low temperature (40 °C) until a semi solid residue was obtained. The crude extract of each part was dried in a vacuum desiccator [14,15].

### 2.2. Qualitative Phytochemical Screening

The crude extract was qualitatively tested for chemical constituents by performing various tests such as Mayer’s test for alkaloid, Salkowski’s test for steroids, aluminum chloride test for flavonoids, and ferric chloride test for phenolic compounds. These were identified by the characteristic color change using the standard tests [16,17,18].

### 2.3. In Vivo Experiment

The adult male Wistar rats (8 weeks old, weighing 180 g) were obtained from Animal Facilities, Veterinary Faculty, Syiah Kuala University, Banda Aceh (Aceh, Indonesia). Animals were used according to the suggested ethical guidelines for the care of laboratory animals, and the experimental protocol used in this study was approved by the Scientific and Ethical Committee of Syiah Kuala University (Ref: 009/KEPH-C/VII/2017, from 28 September 2017). The rats were acclimatized for at least 7 days. They were housed in an animal room with a temperature of 24–26 °C, humidity of 55%–60%, a regular 12 h/12 h light/dark cycle, and standard laboratory administration diet and water were provided ad libitum. General health status of the rats was monitored on alternate days. At the beginning of each experiment, the body weight of the animals ranged from 180 to 220 g. A completely randomized design was used for the experiment, which consisted of 6 treatment groups, each consisting of 4 rats, as follows: 1) NR, normal rat; 2) KN, diabetic rat (negative control); 3) KP, diabetic rats given allopurinol 10 mg/kg body weight; 4) EM150, diabetic rats given the test extract 150 mg/kg body weight/day; 5) EM300, diabetic rats given the test extract 300 mg/kg body weight/day; and 6) EM450, diabetic rats given the test extract 450 mg/kg body weight/day. Diabetes was induced in the animals by intra peritoneal injection of freshly prepared aloxan in a single dose of 150 mg/kg. One week after aloxan injection, fasting blood glucose was measured to verify the development of diabetes. The *M. calabura* L. stem bark extract was given for 45 days. Body weight, naso-anal length, and body mass index were measured as described by Ahmed et al. [19]. The crude ethanol extract from *M. calabura* L. stem bark at 150, 300, and 450 mg/kg were administered orally for 5 days to aloxan-induced diabetic rats, and serum uric acid levels were measured by Gluco check “Nesco”. Blood uric acid level was monitored with the Nesco Multi check (Gesunde Medical, Alamat: Jl. Raya Tajur—Bogor No.256, RT.04/RW.04, Sindangsari, Kec. Bogor Tim., Kota Bogor, Jawa Barat 16146 Indonesia).

### 2.4. Histopathological Determination

For microscopic evaluation, tissues were fixed in a fixative (neutral buffered formalin) and embedded in paraffin, sectioned at 4 μm, and subsequently, stained with hematoxylin–eosin [20,21,22,23]. Sections were studied under a light microscope (DP12 Olympus, Syiah Kuala University, Banda Aceh Indonesia) at 40 magnifications. Slides of all the treated groups were studied and photographed. A minimum of 12 fields of each section was studied and approved by a pathologist who did not know the treatment given. Assessment of kidney damage is done based on the criteria, 0: Normal, 1: Lesions without necrosis, 2: Necrosis [24].

### 2.5. Statistical Data Analysis

Data are expressed as the mean ± standard deviation (SD). Statistical analysis was performed by analysis of variance. Duncan’s multiple range test was performed to determine significant differences. The values of *p* < 0.05 were considered to be statistically significant.

## 3. Results

### 3.1. Phytochemical Screening

Qualitative phytochemical screening of the *M. calabura* stem bark extract shows the presence of flavonoids, alkaloid, triterpenoid, steroid, and poliphenolic compounds.

### 3.2. Effect of Crude Ethanol Extract From M. calabura L. Stem Bark on Body Weight and Body Mass Index

Administration of *M. calabura* L. stem bark extract after 45 days of treatment had an effect on the body weight. Body weight of rats given the stem bark extract was the same as that of normal rats. However, the body weight in diabetic rats was lower when compared with other treatment groups. Administration of *M. calabura* L. stem bark extract had no effect on body mass index (Table 1).

The crude ethanol extract from *Muntingia calabura* L.stem bark at 150, 300, and 450 mg/kg and allopurinol at 10 mg/kg were administrated orally once a day. Values are displayed as mean ± SD. The superscript letters in the same row indicate statistically significant values (*p* < 0.05).

### 3.3. Effect of Crude Ethanol Extract From M. calabura L. Stem Bark on Serum Uric Acid Levels in Rats

The crude ethanol extract from *M. calabura* L. stem bark at 150, 300, and 450 mg/kg were administered orally for five days to aloxan-induced diabetic rats, and serum uric acid levels were measured by Gluco check “Nesco”. As presented in Table 1, compared with the uric acid levels in the normal rats (4.675 mg/dL), the levels in the control diabetic rats (10.225 mg/dL) increased significantly (*p* < 0.05). Administration of the crude ethanol extract from *M. calabura* L. stem bark at 150, 300, and 450 mg/kg could reduce the uric acid levels (*p* < 0.05) compared with the diabetic rats. The positive control, which was administered allopurinol at a dose of 10 mg/kg, displayed hypouricemic activity, which significantly reduced the serum uric acid level, and the same was seen with the rats given the crude ethanol extract from *M. calabura* L. stem bark at 300 and 450 mg/kg (Table 2).

The crude ethanol extract from *Muntingia calabura* L. stem bark at 150, 300, and 450 mg/kg and all opurinol at 10 mg/kg were administrated orally once a day. The control diabetic rat and normal groups were orally administered with aquadest. Values are displayed as mean ± SD. The superscript letters in the same row indicate statistically significant values (*p* < 0.05).

Table 3 showed that in the positive control given allopurinol, there was a histopathological change in the form of necrosis with a percentage of 33%, and after being given *M. calabura* extract of 300 mg and 450 mg, necrosis did not occur. Kidney damage categories can be seen in Table 4.

Data were tested using the Chi square method by looking at the results in the Fisher’s exact test column; because cells had an expected value of less than 5, the *p*-value results used as a hypothesis test are the results of the *p*-value of the Fisher’s exact test and obtained *p* values > 0.05 (*p* = 0.184). This shows that the extract had no significant effect on the histopathological features of the glomerular and renal tubules in experimental animals

## 4. Discussion

The ethanol extract from *M. calabura* L. stem bark at 300 mg/kg displayed the same inhibitory effect on serum uric acid levels as in rats that received allopurinol (Table 2).The presence of flavonoid and poliphenolic compounds in the *M. calabura* stem bark extract may have reduced uric acid in the rats. The polyphenol compound from *M. calabura* stem extract is thought to have an important role in lowering uric acid. Flavonoids have a role in inhibiting the activity of xanthine oxidase. Flavonoids can reduce uric acid by reducing the activity of xanthine oxidase in serum and increasing the concentration of uric acid in urine and binding free radicals as long as the purine changes into uric acid. Flavonoid can inhibit the performance of xanthine oxidase and xanthine dehydrogenase, so that it can inhibit uric acid synthesis [25,26]. In addition, extract of *M calabura* stem bark contain triterpenoid and it was reported that triterpenoid and triterpenoid saponin or riparsaponin could significantly inhibit xanthine oxidase activity so that uric acid decreases in the blood [27,28].

Chen-Yu et al. [27] reported that flavonoids in *Davallia formosana* extract can decrease uric acid levels by inhibition of xanthine oxidase enzyme. Specifically, plant phenolic compounds, such as phenolic acids and flavonoids, show strong antioxidant activity through scavenging free radical. In addition, both types of compounds inhibit xanthine oxidase activity [29,30,31]. Treatment with allopurinol can reduce serum uric acid at the rate of cardiovascular complications in patients with coronary heart disease, congestive heart failure, and dilated cardiomyopathy [29]. The crude ethanol extract of *Siegesbeckia orientalis* displayed antihyperuricemic activity, and the *n*-butanol-soluble fraction was found to be the most active portion of the extract. Further, in vivo studies of this fraction showed 31.4% decrease of serum uric acid levels [14]. Antihyperuricemic effect of novel thiadiazolopyrimidin-5-one analogs has been shown in oxonate-treated rats, which can be further explored not only as antigout therapeutics, but also in other systems where hyperuricemia is the driving cause of the disease [8]. The results indicated that activity of phytochemicals from *D. formosana* significantly inhibited xanthine oxidase activity in vitro and reduced serum uric acid levels in vivo. Activities of flavonoid glycosides could possibly be developed into potential hypouricemic agents [29].

The results of renal histopathology showed that administration of *M. calabura* L. stem bark extract improved the nucleus of proximal renal tubular cells and showed no specific damage to renal proximal tubular cells (Figure 1). In uncomplicated obese, insulin-resistant, and hypertensive patients, serum uric acid levels increase mainly as a consequence of impaired renal excretion, and increased production of uric acid occurs in parallel [32,33]. Body mass index (BMI) has an effect on diabetes. BMI levels were positively correlated with plasminogen-activator network (t-PA Ag), as well as plasminogen activator inhibitor-1 (PAI-1 Ag) concentration, and BMI can be associated with high plasma levels of PAI-1 Ag in Type 2 Diabetes [34].

## 5. Conclusions

It was concluded that *M. calabura* L. stem bark extract decreased uric acid levels, had no specific damage to renal proximal tubular cells, and has potential as antihyperuricemics in diabetic rats. The recommended dose to provide a significant effect on reducing the uric acid level in diabetic rats is 300 mg/kg body weight.

## Figures and Tables

**Figure 1 medicina-55-00695-f001:**
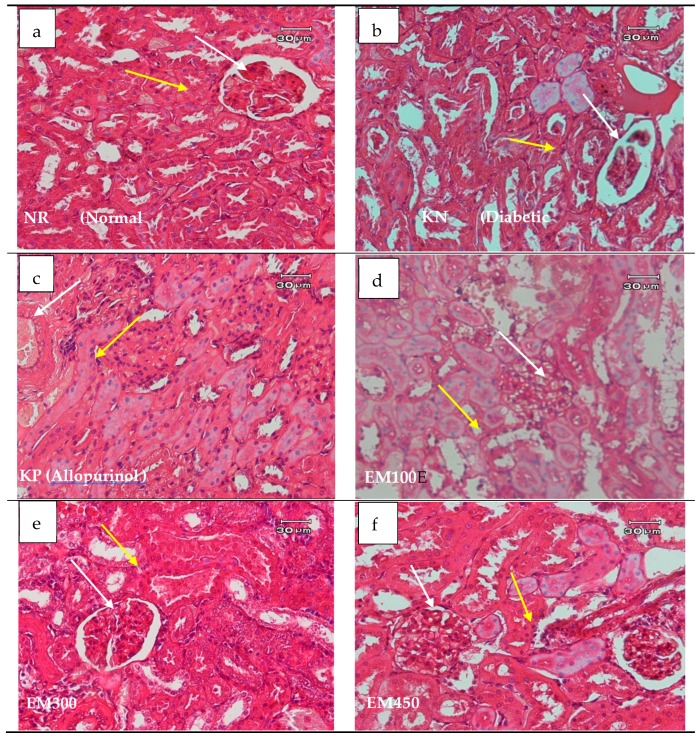
Histopathological studies of the kidney. Histological examination of kidney in rats, which were administered the crude ethanol extract from *Muntingia calabura* L. stem bark at 150 (**c**), 300 (**d**), and 450 mg/kg (**e**) and allopurinol at 10 mg/kg once a day. The control diabetic rat and normal groups (**a**) were orally administered with aquadest, and diabetic rat were aloxan (**b**). Yellow arrow, proximal convoluted tubule; white arrow, glomerulus.

**Table 1 medicina-55-00695-t001:** Body weight and body mass index (BMI) of rats on various treatments.

Treatment	Body Weight (g)	Body Mass Index (g/cm^2^)
NR (Normal rat)	213 ± 6.05^a^	0.44 ± 0.18^a^
KN (Diabetic rat)	169 ± 8.20^d^	0.42 ± 0.02^a^
KP (Allopurinol)	188 ± 9.55^c^	0.47 ± 0.02^a^
EM150	200.5 ± 6.65^b^	0.50 ± 0.01^a^
EM300	208.25 ± 5.5^ab^	0.52 ± 0.01^a^
EM450	201.75 ± 7.13^b^	0.50 ± 0.01^a^

^a,ab,b^ Different superscripts within the same row indicate significantly different (*p* > 0.05).

**Table 2 medicina-55-00695-t002:** Inhibitory effect of crude ethanol extract from *M. calabura* L. stem bark on serum uric acid levels in rats.

Treatment	Serum Uric Acid Levels (mg/dL)	Inhibition (%)
NR (Normal rat)	4.675 ± 2.01^b^	
KN (Diabetic rat)	10.225 ± 1.88^a^	
KP (Allopurinol)	5.000 ± 1.29^b^	51.10
EM150	7.025 ± 3.33^ab^	31.29
EM300	5.125 ± 1.77^b^	50.02
EM450	6.600 ± 2.54^b^	35.45

**Table 3 medicina-55-00695-t003:** Types of histopathological kidney damage to rats.

Treatment	Dv n (%)	Ra n (%)	Rv n (%)	Ri n (%)	Rm n (%)	N n (%)
NR (Normal rat)	1(17%)	0(0%)	1(17%)	3(50%)	2(33%)	0(0%)
KN (Diabetic rat)	1(17%)	0(0%)	4(67%)	3(50%)	3(50%)	3(50%)
KP (Allopurinol)	0(0%)	0(0%)	4(67%)	3(50%)	1(17%)	2(33%)
EM150	1(17%)	0(0%)	1(17%)	2(33%	1(17%)	1(17%)
EM300	0(0%)	0(0%)	1(17%)	2(33%)	1(17%)	0(0%)
EM450	0(0%)	0(0%)	2(33%)	2(33%)	2(33%)	0(0%)

Note: Dv = vacuolar degeneration, Ra = acute inflammation, Rv = inflammation around the vascular, Ri = inflammation in the interstitial, Rm = inflammation in the medulla, N = necrosis.

**Table 4 medicina-55-00695-t004:** Kidney damage category.

Treatment	Number of Samples		
	Normal	Lesions Without Necrosis	Necrosis
NR (Normal rat)	0	4	0
KN (Diabetic rat)	1	4	2
KP (Allopurinol)	1	4	2
EM150	1	3	0
EM300	0	2	0
EM450	0	2	0

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
