# Peer review of "Effect of Muntingia calabura L. Stem Bark Extracts on Uric Acid Concentration and Renal Histopathology in Diabetic Rats"

_medicina, 2019, doi:10.3390/medicina55100695_

Round 1

Reviewer 1 Report

Dear sir, 

thank you to select me to review manuscript Safrida S and Sabri M. Effect of Muntingia calabura L. stem bark extracts on uric acid concentration and renal histopathology in diabetic rats. Authors evaluated natural product of M. calabura L. stem bark extract with allopurinol in diabetic rats with hyperuridemia. The dose of  300 mg/kg body weight had a similar  effect on reducing the uric acid level compared to allopurinol. Design of study is well made, results are interesting, but paper has a several important limitations:

1) Paper needs general English language and style editing, re-evaluation of manuscript is possible only after improvement in English language.

2) Please describe in detail possible effect of  Muntingia calabura L. stem bark extracts on uric acid metabolism

3) Please decribe  the renal histology changes in all groups of rats in detail, add another table evaluating impact of diabetes mellitus and treatment with allopurinol or Muntigia calabura extract on renal histology (part results). Add also electron microscopy findings, if available (authors evaluated effect of treatment on renal tubular pathology)

4) Discussion needs to be more addressed

5) Add limitations of the study, especially small number of evaluated rats.

Major revision of manuscript is needed.

Reviewer 2 Report

The authors investigated the effect of Muntingia calabura L. stem bark extracts on uric acid concentration and renal histopathology in diabetic rats. However, there are some problems had to be detail clarified 

1. Extensive editing of English language is necessary. There are several grammar errors.

Several logic errors were noted in introduction We suggested that the tables were needed to be re-designed again. In the tables, what is the meaning of a, b, c, d??? In the result, the serum uric acid levels in EM 300 group is significant lower than EM 450. Why? Other laboratory data of the rats is important for readers to understand the severity of diabetes, such as TG, LDL, HDL…… How could authors provide the evidence of renal or liver toxicity of M. calabura L. stem bark? The result cannot support the current conclusion. The detail mechanisms were necessary. We suggested the author to provide the detail cell signaling. The information from histological examination of kidney provided less information. We suggested the author to provide the data of IHC stain. It is necessary to report also all the new studies of the literature on this topic which are now missing.  The mechanism of pathophysiological links between M. calabura L.stem bark, DM and urine acid reported in some previous studies. We suggested authors to report current results in article.

Author Response

i

Round 2

Reviewer 1 Report

Dear sir, 

manuscript in revised form could be acceped for publication in Medicina. 

Reviewer 2 Report

We thanks for author to provide detail information in this manuscript  and revised point by point.